# COVID-19 Vaccine Education (CoVE) for Health and Care Workers to Facilitate Global Promotion of the COVID-19 Vaccines

**DOI:** 10.3390/ijerph19020653

**Published:** 2022-01-07

**Authors:** Holly Blake, Aaron Fecowycz, Hollie Starbuck, Wendy Jones

**Affiliations:** 1School of Health Sciences, University of Nottingham, Nottingham NG7 2HA, UK; aaron.fecowycz@nottingham.ac.uk (A.F.); wendy.jones@nottingham.ac.uk (W.J.); 2NIHR Nottingham Biomedical Research Centre, Nottingham NG7 2UH, UK; 3High Wycombe Campus, School of Nursing, Midwifery and Allied Health, Buckinghamshire New University, Buckinghamshire HP11 2JZ, UK; hollie.starbuck@bucks.ac.uk

**Keywords:** COVID-19, vaccine, healthcare, social care, digital, health education, health protection

## Abstract

The COVID-19 vaccine is being rolled out globally. High and ongoing public uptake of the vaccine relies on health and social care professionals having the knowledge and confidence to actively and effectively advocate it. An internationally relevant, interactive multimedia training resource called COVID-19 Vaccine Education (CoVE) was developed using ASPIRE methodology. This rigorous six-step process included: (1) establishing the aims, (2) storyboarding and co-design, (3) populating and producing, (4) implementation, (5) release, and (6) mixed-methods evaluation aligned with the New World Kirkpatrick Model. Two synchronous consultations with members of the target audience identified the support need and established the key aim (Step 1: 2 groups: *n* = 48). Asynchronous storyboarding was used to co-construct the content, ordering, presentation, and interactive elements (Step 2: *n* = 14). Iterative two-stage peer review was undertaken of content and technical presentation (Step 3: *n* = 23). The final resource was released in June 2021 (Step 4: >3653 views). Evaluation with health and social care professionals from 26 countries (survey, *n* = 162; qualitative interviews, *n* = 15) established that CoVE has high satisfaction, usability, and relevance to the target audience. Engagement with CoVE increased participants’ knowledge and confidence relating to vaccine promotion and facilitated vaccine-promoting behaviours and vaccine uptake. The CoVE digital training package is open access and provides a valuable mechanism for supporting health and care professionals in promoting COVID-19 vaccination uptake.

## 1. Introduction

The World Health Organization (WHO) declared the outbreak of coronavirus disease (COVID-19) a pandemic in March 2020. COVID-19 is caused by the severe acute respiratory syndrome coronavirus 2 (SARS-CoV-2). As of June 2021, there were over 176 million cases, and 3.82 million confirmed deaths attributed to COVID-19 worldwide [1]. In response to its high mortality and rapid spread, new vaccines have been developed and tested at an unprecedented pace, described as the ‘prime weapon’ in the fight against escalating daily death rates [2].

The success of COVID-19 vaccination programmes relies not only on high population coverage, but also on high rates of acceptance amongst the general public and healthcare workers. A recent systematic review including studies from 33 countries showed that vaccine acceptance is highly variable, ranging from 23.6% to 97% in the general public [3]. Other systematic reviews and meta-analyses showed that rates of vaccine acceptance and intention to vaccinate declined during 2020 [4,5], with evident social inequalities in vaccine hesitancy [4]. The most frequently raised concerns are related to side effects of the vaccines, and a belief that the vaccines were not sufficiently tested [6].

Although trust in the vaccines has been climbing in 2021, there is a need to reassure the public about the importance of the COVID-19 vaccine, and its safety and effectiveness [6]. Healthcare professionals are a trusted and credible source of vaccine-related information [7,8] and play an important role in dispelling myths about vaccines and building public confidence in vaccination. They have a powerful influence over vaccination decisions of members of the public [9]. However, vaccine acceptance in healthcare workers (HCWs) is also variable, ranging from 21% to 71.8% [3]. COVID-19 vaccine hesitation has been identified among HCW in many countries [10,11,12,13,14,15,16]. There is variation in vaccine acceptance and uptake between occupational groups [17,18], and ethnic minority HCWs are less likely to take up vaccination [4,18,19]. Although rates vary across countries, one recent survey (Libya, *n =* 15,087) found that only 14.9% of respondents believed that vaccination benefits outweighed the risks [20]. This is important since health professionals are more likely to recommend vaccination if they themselves have been vaccinated [21] and people are more willing to receive the vaccine if a healthcare provider recommends it [22]. HCWs with less confidence in the benefits and safety of vaccines are less likely to recommend vaccines to patients and their families [23,24,25,26].

Behavioural research has shown that, beyond creating an enabling environment, vaccine acceptance and uptake can be increased by harnessing social influences and increasing motivation [7]. Leveraging the role of HCWs is one approach to harnessing social influences. Vaccination decision-making is influenced by HCWs [22], and vaccine acceptance is known to be associated with greater COVID-19 knowledge [27]. Therefore, improving HCWs’ knowledge about the COVID-19 vaccine, and providing evidence-based tools to support their promotion of vaccination, could lead to greater vaccine uptake. Educating healthcare professionals about the risk of COVID-19, efficacy of the vaccine and tackling disinformation is crucial to increasing vaccine uptake globally [28]; and in HCWs, healthcare students, and the general public by maximising opportunities for validation, endorsement, or persuasion [7]. Anecdotally, healthcare students and healthcare professionals who are not trained vaccinators have reported feeling ill-equipped to advocate the COVID-19 vaccine to patients, clients, and the public, or to answer their questions about the value of the vaccine, its safety, and effectiveness. To better equip HCWs with the knowledge and skills to increase peoples’ motivation to vaccinate, educational interventions for HCWs should address motivational barriers such as low perceived risk and severity of infection, fear, worry, and low confidence in vaccines [7]. Low confidence in vaccination may result from a lack of knowledge about effectiveness, concerns about side effects, influence of religious values, and exposure to misinformation, conspiracy theories, and rumours [7].

Since COVID-19 and its associated vaccine only emerged very recently, healthcare curricula have not previously incorporated education on this subject and so the subject area is relatively new for many healthcare professionals and healthcare trainees who are not trained COVID-19 vaccinators. Healthcare professionals, healthcare educators, and healthcare trainees hold positive attitudes towards online learning [29] and digital approaches to learning are now mainstreaming in health education [30]. Advantages of online learning include flexibility, self-pacing, catering to different learning styles and reducing resource costs associated with time, travel, and trainer availability [31,32,33,34]. With the urgency of COVID-19 vaccine (including booster vaccine) rollout globally, the overall aim of this study was to rapidly develop and test an internationally relevant, multimedia e-learning package providing education about the COVID-19 vaccine for health and care workers (and trainees), in order to facilitate global promotion and uptake of the COVID-19 vaccines. The research question was: Does this digital training package improve users’ knowledge and confidence for promoting the COVID-19 vaccine and/or lead to changes in behaviour around vaccine promotion?

## 2. Materials and Methods

A reusable learning object (RLO) was developed, released, and evaluated using ASPIRE methodology [35], drawing on Kirkpatrick Foundational Principles and Kirkpatrick levels of training evaluation [36,37]. The development process was undertaken rapidly during a 6-week period in March–May 2020 to ensure that the resulting RLO would be timely for distribution during the COVID-19 pandemic and rollout of vaccination worldwide.

### 2.1. Reusable Learning Objects

RLOs are short, self-contained, multimedia web-based resources including audio, text, images, and/or video and which engage the learner in interactive learning through the use of activities and assessments [38] towards a single learning objective or goal. They typically have four components: Presentation of the concept, fact, process, principle, or procedure to be understood by the learner in order to support the learning goal. An activity: something the learner must do to engage with the content to improve understanding. A self-assessment: a way in which the learner can apply their understanding and test their mastery of the content. Links and resources: external resources to reinforce the taught concept and support the learning goal [39]. The reusability of a RLO is established through licensing models such as a Creative Commons License, which allows the owner of the material to distribute RLOs freely for use whilst retaining the ownership.

### 2.2. ASPIRE Methodology

A rigorous design and development approach for materials is important: a clear, simple, and consistent conceptual model will increase the usability of a system [40] and support widespread uptake and use of digital resources. Therefore, development and testing of the RLO was undertaken using ASPIRE methodology (Figure 1) which is a well-used and validated tool for RLO development [35] and an approach that is suggested to fit optimally with requirements for designing high quality digital training in healthcare [41]. ASPIRE methodology uses participatory co-design principles and is centred on developing a ‘community of practice’ [42] of experts and potential future users who work together at each stage of the process, to identify learning needs and create content supported by instructional designers and multi-media developers.

ASPIRE has a six-step process [35,43] including: (1) establishing the aims of the RLO, (2) storyboarding, (3) populating/production, (4) integration, (5) release, and (6) evaluation, which we aligned to the New World Kirkpatrick Model [36,37,44]. ‘Aims’ refers to the need to have a clear focus for the resource. This includes the topic area to be covered or learning goal, and the characteristics of the target group of learners. ‘Storyboarding’ is where stakeholders come together to work creatively on ideas for the content and design of the resource using storyboards. ‘Populate and Produce’ is where the ideas are translated into media components ready to be ‘integrated’ together using a suitable platform such as HTML5. ‘Release’ relates to how the resource will be made available to learners via a virtual learning environment (VLE), repository, or website, for example, and how will it be promoted. The final stage, ‘evaluate’ is determining the efficacy of the learning resource in a real learning situation.

Two synchronous consultations with members of the target audience identified the support need and established the key aim for the RLO (Step 1, 2 groups: *n* = 48). Asynchronous storyboarding was used to co-construct the content, ordering, presentation, and interactive elements (Step 2: *n* = 14). The project team populated the content template and produced relevant graphics and media (Step 3, *n* = 3). This was integrated into a RLO template through a technical development process. Both content (specification) and technical (media) development were undertaken by the project team (*n* = 3). Two-stage peer review of content and technical presentation was undertaken (Step 3: *n* = 23). The final RLO was uploaded to HELM Open (https://www.nottingham.ac.uk/helmopen/ (accessed 17 December 2021)) and released in May 2021, with evaluation data collected via an embedded survey, and post-training qualitative interviews. The process is shown in Figure 1, and details for each step are described below.

#### 2.2.1. Step 1: Establishing the Aims

Two synchronous group consultations were undertaken in February 2020 with healthcare professionals (Group 1) and healthcare students (Group 2) in the UK, aligned with scheduled public health education and training sessions. The purpose was to establish the topic area to be covered and the learning outcomes, and the characteristics of the target audience. The focus of these consultations was to explore participants’ views towards, and knowledge of, the COVID-19 vaccine and to discuss barriers and challenges in communicating with patients and clients about vaccination with intention to encourage vaccine uptake. The consultations were led by a health psychologist and health educator, who delivered a 20-min introductory presentation on ‘Public health and vaccines’, followed by a 40-min group discussion. The two sessions were held remotely using Microsoft Teams (Redmond, WA, USA).

Group 1 included 28 nurses, who had been registered between 2 and 40 years (86% female, 32% delivering vaccinations; 25% from ethnic minority groups). Group 2 included 20 healthcare students aged 18–42 years (55% female; 10% delivering vaccines as registered nurses, 65% from ethnic minority groups). Ethnic minority groups were purposely over-sampled due to the variations in the prevalence of vaccine hesitancy, since vaccine uptake (for COVID-19 and previous national vaccination programmes in the UK) has been lower in areas with a higher proportion of minority ethnic groups. Additionally, ethnic minority HCWs are less likely to take up vaccination themselves [4,18,19]. The views of ethnic minorities were therefore essential in this study. The proportion of attendees from ethnic minority groups was higher than the proportion of minority groups in the general UK population (13%) [45] and the UK National Health Service (NHS) (22.1%) [46] and was higher in Group 2 than the proportion of ‘non-White only’ people within the general US population (33.7%) [47] and US healthcare workforce (35.6%) [48].

The key points for each discussion group were:Would you have the COVID-19 vaccine if it was offered to you?Would you encourage others to have the COVID-19 vaccine? Why?How confident do you feel in your ability to communicate with patients, clients, or the general public about the COVID-19 vaccine?Are there any barriers and challenges to effective communication about the vaccine?

The vast majority of participants were highly positive towards vaccination and believed that it was important for healthcare professionals to encourage uptake of the COVID-19 vaccine, and other vaccines in general. Many participants indicated that they would take (or had taken) the vaccine themselves and would encourage (or had) their families to vaccinate. However, 25% (5/20) of the healthcare students and 14% (4/28) of the healthcare professionals reported that they would not personally take the vaccine or advise family members to do so. Of the 9 participants who were hesitant to vaccinate, 8 were from ethnic minority groups and they highlighted the speed of development of COVID-19 vaccines, concerns about contracting COVID-19 from the vaccine, and discussed rumours about the vaccine’s purpose and possible side effects circulating within their community groups. These views dominated conversations and led to some group members who were initially positive about the vaccine being unable to respond or doubting their initial response or own knowledge. Most of the healthcare students and many of the healthcare professionals reported having low confidence in their ability to describe key facts relating to the COVID-19 vaccine and to respond to questions from others.

Many participants (particularly those who were less knowledgeable about the vaccine) felt that access to evidence-based information would increase their knowledge and confidence to promote uptake of the COVID-19 vaccine and improve their ability to discuss the vaccine with, and answer questions from others (Figure 2). It was perceived that the volume of online information in public-facing websites was overwhelming and needed to be more digestible. It was unanimously agreed that a digital resource, such as an e-learning package, would be the most appropriate format.

Based on the group discussions and expertise within the project team, the agreed aim of the e-learning resource was to ‘increase understanding about the COVID-19 vaccine and provide a resource that will help healthcare professionals and healthcare students to explain to patients and clients why vaccine uptake is important for individual and societal health’.

#### 2.2.2. Step 2: Storyboarding

In this step, the content for the RLO was drafted, through a process called ‘storyboarding’. A rapid storyboarding exercise was undertaken over a period of one week, with a group of 14 healthcare professionals and members of the public, to establish the key messages, content, and design for the RLO. Due to the urgency of the COVID-19 pandemic situation, the storyboarding was asynchronous (conducted virtually, using prepared resources and without real-time facilitator interaction). This was to ensure that all participants could contribute within a short timescale, with the overall aim of developing a timely and high-quality output that would be of genuine value to health and social care organisations during a global pandemic situation. Individuals were purposively selected via professional networks to ensure participants represented the views of those with knowledge of vaccination programme delivery in different contexts and settings, vaccination uptake and decision-making; the group included 2 medical doctors, 2 health psychologists, 5 nurses, 2 occupational health specialists, and 3 members of the general public. This group constituted an expert ‘community of practice’ to assist in refining the storyboard.

The questions put to the group were:What are the major areas to be covered?What is the best sequence and structure for the material?What do you want the users to be doing at each stage of the process?How will users assess whether they have achieved the learning goal?

The storyboarding activities resulted in a final contents list (Figure 3) and framework for the resource specification, with agreement on ordering, presentation, and the use of interactive elements. The agreed learning outcome was to understand the importance of the COVID-19 vaccine for individual and societal health. Although RLOs are designed to address a single learning outcome, it was agreed that the release of the resource during a global pandemic meant that additional information was needed to ensure that learners fully understood the need to promote adherence to behavioural measures concurrently with vaccine uptake, and to carefully consider mechanisms for promoting vaccine uptake. Therefore, the agreed key message of the RLO content was that ‘a COVID-19 vaccine, when used in combination with current public health measures such as physical distancing, face masks, respiratory etiquette, and hand hygiene has the potential to reduce the significant burden of COVID-19′. It was proposed that general information about the value of vaccines was required to set the context prior to presenting materials on COVID-19 and the COVID-19 vaccine. It was also agreed that the content should communicate that the evidence situation is evolving with relation to COVID-19 and the vaccines. Following the storyboarding exercise, the initial full content draft was co-created by the project team. The project team consisted of a health psychologist, an occupational health nurse, and a learning technologist.

#### 2.2.3. Step 3: Populate and Produce

The content template was then populated by the project team. Interactive images, information buttons, a quiz, and reflection through a feedback survey were included. Pedagogical design principles for multimedia learning were adopted from Wharrad and colleagues [41] to translate ideas into media components (Table 1).

The technical presentation of the RLO allowed for the user to adapt the type of media used to deliver the content, to allow reduction in cognitive load, to maximise accessibility and to improve learning experience. For example, the learner might switch text and audio on or off, and control media elements, by pausing video or slowing down the audio narration. A user can therefore decide how the information is delivered and so the RLO is adaptable to different contexts and devices.

Once the proposed RLO was complete, an international peer review panel of 23 experts was established. Panel members were purposively selected via professional networks to ensure participants represented a range of health and social care disciplines, levels of seniority, and settings. They had expertise in health and medical education, public health strategy, virology, biology, medicine, nursing and allied health, pharmacy, health psychology, sociology, and occupational health. Reviewers included COVID-19 vaccinators and experts in digital health communications and design. Reviewers were from seven countries (United Kingdom, United States of America, Pakistan, Jordan, Turkey, Thailand, and Malawi) to establish the relevance and appropriateness of content across a range of cultures and geographical regions. The review panel completed standard Stage 1 specification review forms accessed from HELM Open (Appendix A).

#### 2.2.4. Step 4: Integration

The media components of the RLO were integrated using a bespoke, accessible, and user-friendly HTML5 template which embraces a mobile-first design philosophy. The template ensures the best possible user experience whatever device is being used to access the resource. This technical development stage was undertaken by a learning technologist from the Health E-Learning and Media Team (HELM) at the University of Nottingham working together with the project team. To evaluate this, the expert review panel completed standard Stage 2 media review forms accessed from HELM Open (Appendix A). Taking a pragmatic approach in the context of a pandemic, expert peer review of both specification and media aspects was undertaken concurrently, and the final resource was also tested for understandability and functionality with 5 members of the general public. Iterative review of the resource by all project team members continued throughout the process. The key revisions and overall findings from the peer review process are shown in Figure 4.

The outcome of content and technical development is shown in Figure 5 (screen examples). The final version of the RLO included audio narration and allowed users to download a certificate of completion.

#### 2.2.5. Step 5: Release

The final RLO was uploaded to HELM Open, a repository of over 200 freely available RLOs at the University of Nottingham. It was released as an open access resource on 3rd June 2021 at the following URL: https://www.nottingham.ac.uk/helmopen/rlos/practice-learning/public-health/CoVE/, version 1.0, accessed on 17 December 2021) and made available to users by circulating through professional networks and social media.

#### 2.2.6. Step 6: Evaluation

Evaluation aligned with the four levels of the New World Kirkpatrick Model [36,37,44], which is a widely used approach to analysing and evaluating the results of training and educational programs. The aim of the evaluation was to explore the perceived influence of CoVE training for health and social care professionals, through:

(i) Level 1: determining user reaction. This is reflected in the degree to which participants found CoVE training favourable, engaging and relevant to them, and/or their job role.

(ii) Level 2: establishing new learning. This is reflected in the degree to which participants acquired knowledge (‘I know it’), skills (‘I can do it right now’), attitude (‘I believe promoting the vaccine is worthwhile’), confidence (‘I think I can promote the vaccine’), and commitment (‘I intend to promote the vaccine’) based on participation in CoVE training.

(iii) Level 3: describing knowledge transfer/behaviour. This is reflected in the degree to which participants applied what they learned from the CoVE package in their job role or daily lives (behaviour change). Required drivers for behaviour change include any processes and systems that reinforced, encouraged, or rewarded promotion of the COVID-19 vaccine.

(iv) Level 4: exploring results and impact. This is reflected in the degree to which targeted outcomes occurred as a result of CoVE training. Leading indicators for this impact included observations that critical behaviours were on track to create a positive impact on desired results.

Table 2 shows a mapping of data collection approaches for each sub-component within the four Kirkpatrick levels. The evaluation took place over an 8-week period July-Sept 2021. Data were collected using pre (2 items) and post (14 items) survey questions embedded within the e-learning package, and a post-exposure interview. Survey items (Appendix A) were adapted from the ‘Evaluation Toolkit for Reusable Learning Objects and deployment of e-Learning Resources’ [49]. Post-exposure semi-structured interviews were conducted by an independent researcher who had not been involved in the RLO design or development. Interview participants were recruited through health and social care professional networks and promotional mailings. Female participants were purposely over-sampled. This was to reflect the gender balance in health and social care (70% female, across 104 countries [50]). Potential participants were provided with a link to Jisc Online survey, where they could access the participant information sheet, and provide online consent to take part. Interviews took place shortly after participants had completed the training package (within four weeks) lasted between 12 and 36 min (average 18 min) and followed a topic guide (Appendix A). They were conducted remotely by telephone or Microsoft Teams, audio-recorded with consent, and were fully transcribed. Analysis followed principles of framework analysis [51] to allow insights from the interview data to be mapped directly to the Kirkpatrick Evaluation Framework [36,37]. On the basis of the researchers’ prior experience in the development and evaluation of digital training packages, and qualitative samples in published evaluations of RLOs or participants’ views towards them [52,53,54,55] (*n* = 6–15), we estimated that recruitment of 12–15 interview participants would achieve sufficient information power to map data to the pre-defined criteria of the framework and meet the study objectives. The study protocol was considered exempt from full research ethics review by the University of Nottingham Faculty of Medicine and Health Sciences Research Ethics Committee in July 2021 (Ref: FMHS 310-0721).

## 3. Results

Mixed-methods analysis aligned with the New World Kirkpatrick Evaluation Framework [36,37] is provided in Table 3. This includes data from 162 online survey participants, and qualitative interviews conducted with 15 participants, (13 health or social care professionals, 3 students; 1 held both roles). Interview participants were nurses (*n* = 12), social scientists (*n* = 2), occupational health specialists (*n* = 1) and COVID-19 vaccinators (*n* = 2), who identified as British, Filipino, Polish, Lebanese, and Pakistani.

### 3.1. Level 1 (Reaction: Reach, Use, Satisfaction, Engagement, Relevance)

CoVE had wide reach, with participants from 26 countries: Algeria, Australia, England, Finland, Ghana, Greece, Guernsey, France, Ireland, India, Indonesia, Italy, Jordan, Lebanon, Malawi, Nigeria, Pakistan, Philippines, Poland, Romania, Scotland, South Africa, Thailand, Uganda, United States of America, Wales. Participants had mostly accessed the package through employers (public and private hospitals, public health or clinical commissioning groups, family doctors, local government networks), professional networks, charitable or volunteering organisations, and higher education institutions. Most survey participants were health and social care professionals (and trainees), or public health specialists. The main reasons for accessing CoVE and the elements of the package that were most valued are presented in Figure 6. Almost all participants found CoVE easy to use and helpful, reported high satisfaction with the training and would recommend it to others. Recommendations for improvement were few and related mainly to the inclusion of additional detail (which was beyond the scope of the learning objective or was already included in additional resources). A small number of users highlighted a need for additional material to meet specific needs of their culture or region, although there were no barriers raised to use of the existing material. Technical issues were few and mostly related to issues with individual devices or internet access. Participants were highly engaged in the package—there were 3653 page views during the data collection period; records only include individuals who consented to web analytics tracking and so the actual engagement figure is likely to be significantly higher. Overall, the content was perceived to be highly relevant across health and social care professions, and diverse geographical regions.

### 3.2. Level 2 (Learning: Knowledge, Skills, Attitudes, Confidence, Commitment)

Following exposure to CoVE there was a significant increase in the proportion of participants who rated their knowledge level as 8/10 or higher (pre-survey: 35.5%; post-survey 84.6%). Most participants reported increased skills to facilitate conversations with others about the COVID-19 vaccine and respond appropriately to questions, particularly from individuals who were more hesitant towards vaccination. There was evidence of change in attitudes towards the vaccine. For those with existing positive attitudes, their views had been consolidated by the evidence-based materials. However, some participants spoke of their own hesitancy towards the COVID-19 vaccine (e.g., general worries about vaccines, or specific concerns about the speed of vaccine development) but noted that their concerns had been allayed after engaging with the package. Most of the participants felt that their confidence in promoting vaccine uptake had improved. Participants believed that the package had helped them to communicate more effectively about the COVID-19 vaccine with diverse audiences, including patients and clients, healthcare students, peers, and the general public, including their own family members. They referred to having increased confidence that they could present the facts (including benefits and risks), while dispelling myths and rumours. While this view was common across the sample, increased confidence was particularly notable in health and care professionals who were working in areas with high levels of vaccine hesitancy, and/or low vaccine uptake rates. Many of the participants demonstrated a commitment to adoption of CoVE within their setting that they believed would have a future impact, and some had already made firm plans to do so. Beyond personal use of the materials, participants intended to share the materials with others (work colleagues, professional networks, family), use the package for continuing professional development training within their teams, and incorporate the materials into new staff inductions (e.g., in care homes).

### 3.3. Level 3 (Transfer/Behaviour and Required Drivers)

While most of the participants reported commitment and future intentions, many participants had already enacted changes in their own vaccine promoting behaviour, as well as supporting their peers with the same. Many of the participants had subsequently engaged in conversations with others about the COVID-19 vaccine and felt that they were able to do this more effectively with their newfound knowledge and confidence. Participants proposed a range of required drivers for knowledge transfer and effecting behavioural change. It was proposed that CoVE training could be targeted to specific professional groups who had high levels of patient contact (e.g., nurses, healthcare assistants, healthcare students, and community pharmacists), or in specific settings with lower vaccine uptake rates, lower levels of knowledge and awareness, and greater vaccine hesitancy (e.g., specific geographical regions, community or ethnic groups, or settings such as care homes). Centralising access was proposed as a mechanism for wider distribution (and resulting behaviour change), for example, through higher education settings, professional networks, or governments. While the digital presentation was unanimously positively received, one participant suggested that a paper-based format may help to widen access (e.g., in rural areas with lower levels of internet access, and fewer people with access to electronic devices).

### 3.4. Level 4 (Impact)

Participants reported positive impacts of CoVE on vaccination uptake. Since accessing CoVE, a few participants shared that they had personally been vaccine hesitant and had re-considered their own decision not to vaccinate following use of the package. Others believed that the knowledge and confidence they had gained from using CoVE had facilitated their discussions about vaccination with vaccine-hesitant individuals who had subsequently vaccinated. Participants reported positive outcomes for vaccination uptake with relation to their peers (health and care professionals), patients, and family members. There were many leading indicators of future impact. For example, participants had used their newfound knowledge and confidence to engage in individually focused vaccination-promoting activities and had been successful in changing people’s attitudes towards the COVID-19 vaccine to reduce vaccine hesitancy and encourage future uptake. Others had utilised the package for wider knowledge-exchange activities, such as the establishment of COVID-19 awareness events and new vaccination programmes, and provision of training for health and care staff, or students.

## 4. Discussion

The aim of the study was to rapidly and rigorously develop and test a multi-media digital package providing timely education about the COVID-19 vaccine for health and care workers. The CoVE package was developed and tested in collaboration with international experts, using ASPIRE methodology [35] with a mixed-method evaluation mapped to the New World Kirkpatrick Model [36,37]. To our knowledge and at the time of writing, CoVE is the first internationally relevant digital training resource for health and care professionals, focused on advocating the importance of the COVID-19 vaccine to individual and societal health.

The package has global reach and was positively received by users from 26 countries, spanning geographical regions with higher and lower rates of vaccination at the time of the study [56]. CoVE was perceived to be timely, and relevant across professions, cultural groups, sectors, and occupations (health and social care, education, charitable and volunteering organisations, local government).

Package users were highly engaged with the materials. Digital learning is known to be enhanced by multimedia [57,58] and the RLO therefore included a mix of text, embedded video, and audio narration which was well-received. Interactivity is recognised as a key method for increasing knowledge through e-learning by keeping users active in the process [59,60]. In this study, the high level of engagement translated into knowledge acquisition and a perceived increase in skills and confidence for promoting vaccine uptake. As a result, package users reported or observed positive changes in attitudes towards the COVID-19 vaccine. This is important since positive attitudes towards vaccination (the ‘proximate behaviour’) have been linked to vaccination uptake (the ‘ultimate behaviour’) [61] which has been demonstrated in other vaccination contexts (e.g., childhood vaccinations [62]).

Participants in CoVE training reported increased confidence and skills and feeling better equipped to communicate with others about the COVID-19 vaccines. Many found the package useful in helping them to correct misinformation (which is known to be associated with changing health beliefs) [63]. Their increased confidence translated into behavioural changes related to engagement in health protection activities (e.g., peer-to-peer training, student education, organisation of vaccination advocacy events, supporting COVID-19 vaccination programmes). The immediate impact of CoVE was evident in reports of vaccination uptake (e.g., themselves, their peers or family members, or members of the general public they had communicated with).

Although the package was valued by users irrespective of their job role or country of origin, changes in confidence and vaccine-promoting behaviour were particularly evident in those health or social care professionals who were not trained vaccinators, or were from geographical areas which have seen lower rates of COVID-19 vaccination [64], and/or high levels of vaccine hesitancy (e.g., [3,10,65,66]). Vaccine hesitancy refers to adults who have been offered the coronavirus vaccine and have chosen not to be vaccinated or report being unlikely to have the coronavirus vaccine if it is offered to them [67]. The causes of vaccine hesitancy are complex and context specific [68]. While CoVE training includes information on development, safety and effectiveness, its primary learning objective relates to the importance of the COVID-19 vaccine for individual and societal health. This was deemed to be an appropriate focus since prior research (using data from 149 countries and 284,381 individuals) has shown that confidence in the importance of vaccines (rather than in their safety or effectiveness) had the strongest univariate association with vaccine uptake compared with other determinants considered [69]. Nevertheless, since health workers are the most trusted sources of guidance about COVID-19 vaccines, messages included within CoVE highlighting vaccine efficacy and safety, when delivered by healthcare workers, may be an effective approach for addressing any vaccine hesitancy [70].

Due to the rapid nature of the study, in the context of global rollout of COVID-19 vaccination programmes, the evaluation data were collected immediately after participants had accessed CoVE. As such, the longer-term impact of this training on vaccination outcomes is not yet known. Nevertheless, even in the short timescale, there was clear evidence of behaviour change in terms of participants reporting marked changes in their confidence to promote the vaccine, or immediate changes in their own actions around health protection (i.e., COVID-19 vaccine advocacy work). Users had applied their learning in practice and also demonstrated a high level of commitment to future use of the resource, resource sharing, and application of knowledge and skills. Interview participants identified the key drivers of future change primarily with relation to occupational groups that could be targeted for distribution of CoVE training (e.g., nurses, healthcare assistants, pharmacists, healthcare students), and routes for embedding training (e.g., staff inductions, continuing professional development programmes in health and social care organisations or through professional networks, higher education curricula). The training package takes a proactive approach to vaccine education by targeting training to health and care professionals for three reasons. First, the confidence of those who interface with patients in a clinical setting is critical for presenting a unified message of vaccine support in the medical community [71]. Second, people’s trust in messages communicated by the health and medical community is a key determinant of vaccine uptake [72]. Third, the beliefs and actions of health professionals can influence vaccine uptake in hesitant others [73].

### Study Strengths and Limitations

The evaluation was mapped to specific indicators on the New World Kirkpatrick Evaluation Model [36] as a theoretical framework and focuses solely on these outcomes. The model has been used to evaluate technology-based interventions in a range of other health contexts [74,75,76]. While few studies adopting this model assess outcomes at all four levels, the meaningfulness of the findings is maximised here through assessment of levels 1–4, allowing for the study to demonstrate the Return on Expectations (RoE) of CoVE as a digital training package. In this context, the RoE is the perceived value of the package in improving knowledge, confidence and intended or actual behaviour change related to promotion of the COVID-19 vaccine. The timescale from package exposure to interview data collection varied between participants up to four weeks, due to the need for rapid completion of the evaluation. This was to ensure that the findings would be available (and therefore timely and relevant) within the changing context of a global pandemic and rapid rollout of vaccination programmes worldwide. This meant that some interview participants had more time to reflect on, or act on, information in the package than others. Further research is needed to explore in more depth any influence of the training on intended and actual behaviour change (levels 3 and 4 of the model), especially given the escalation in and varying COVID-19 vaccine information that individuals obtain from friends, family, community groups, the media, and social network platforms. The pre-post assessment of knowledge was based on participants’ self-rating of knowledge (and knowledge change), and so we do not know whether objective knowledge changed. However, this approach was intentional, to allow for ‘ipsative assessment’ in which participants measured their current perceived knowledge and confidence level (on completion of the training) against their perceived knowledge and confidence level at some point in the past (immediately prior to completing the training), to allow for a self-assessment of improvement. This is pertinent since perceived knowledge (and not just factual knowledge) can increase one’s confidence in their own skills [77].

We did not collect sociodemographic information from survey participants (e.g., age, gender, ethnic minority status, religious group, socio-economic status, and underlying health conditions), so the potential influence of these factors on our participants’ views is not known. However, the content of the CoVE package addressed all of the causal factors for hesitancy—including mistrust, concerns around safety and future health risks, (mis)information, societal attitudes, beliefs, and values [78]. The sample of survey participants (*n* = 162) was adequate to address the study objectives and comparable with, or larger than samples in other evaluations of digital training for health and care workers [29,79]. However, evaluation with a larger cross-country sample (or studies with a focus on specific cultural groups) would be valuable to further establish whether the CoVE training represents the specific needs of diverse cultures and regions. The approach to sampling for the evaluation interviews through professional network and promotional mailings allowed participants to self-nominate to participate, thus reducing potential for selection bias. However, it is possible that healthcare workers opting to participate may have a more favourable view towards either COVID-19 vaccination, or digital approaches to education and training.

## 5. Conclusions

The COVID-19 Vaccine Education (CoVE) training package increases users’ knowledge and confidence in communicating with patients, clients and the general public about the importance of the COVID-19 vaccine for individual and societal health. CoVE is internationally relevant, and timely for distribution to health and care professionals and healthcare trainees during the COVID-19 pandemic. We recommend that healthcare organisations and educational facilities widely distribute CoVE to facilitate global promotion and uptake of the COVID-19 vaccines. While CoVE has shown to be globally relevant and provides a wealth of additional evidence-based resources, in certain contexts the training could be delivered alongside additional materials that are tailored to the concerns of motivations of specific cultural groups, or the package could be distributed by trusted members of community groups. The package content has high value at the time of this study but will need to be periodically reviewed and updated. This is because the pandemic’s trajectory (and the response to it) will evolve, vaccines will be more widely distributed, the extended period of media coverage may raise additional questions, and post-vaccination surveillance data will provide greater insights over time.

## Figures and Tables

**Figure 1 ijerph-19-00653-f001:**
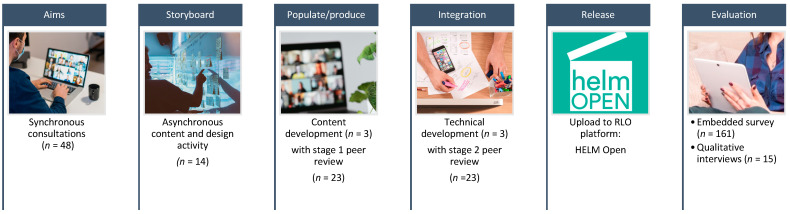
ASPIRE methodology for CoVE package development.

**Figure 2 ijerph-19-00653-f002:**
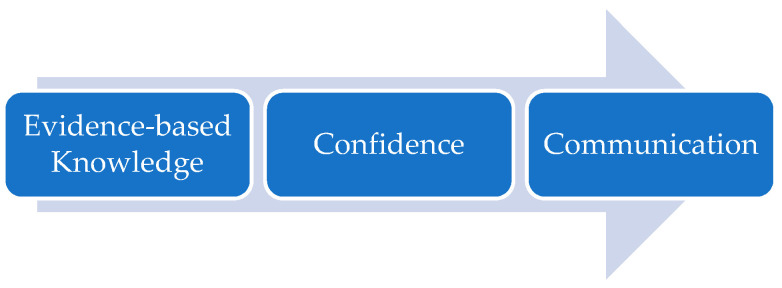
Core outcome of Step 1.

**Figure 3 ijerph-19-00653-f003:**
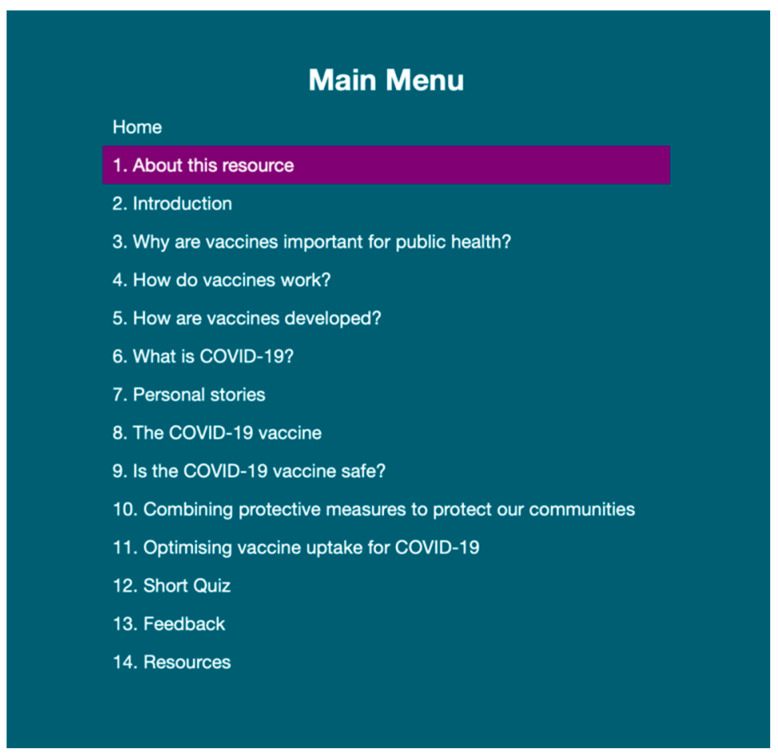
Final RLO contents list resulting from asynchronous storyboarding.

**Figure 4 ijerph-19-00653-f004:**
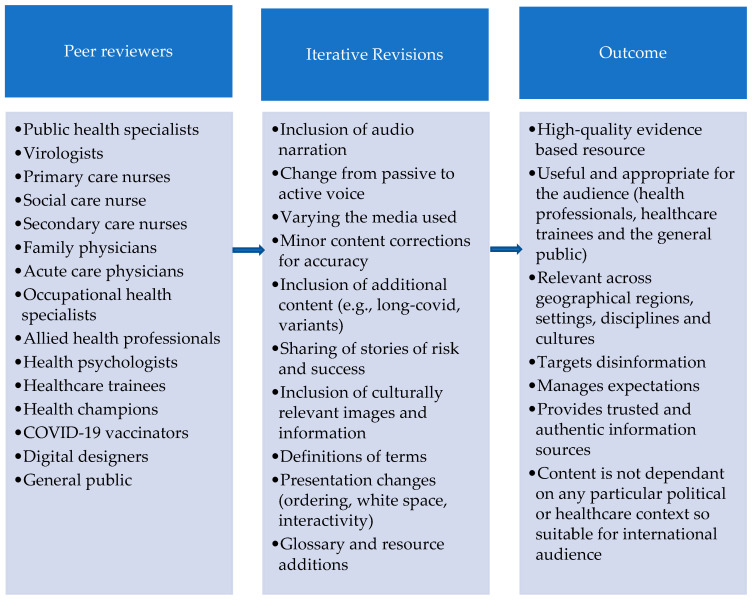
Co-design through expert and lay peer review.

**Figure 5 ijerph-19-00653-f005:**
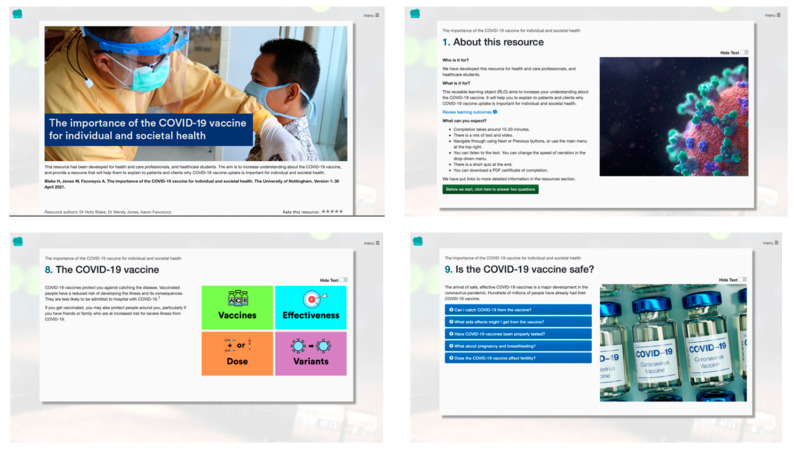
Screen examples from the final developed RLO.

**Figure 6 ijerph-19-00653-f006:**
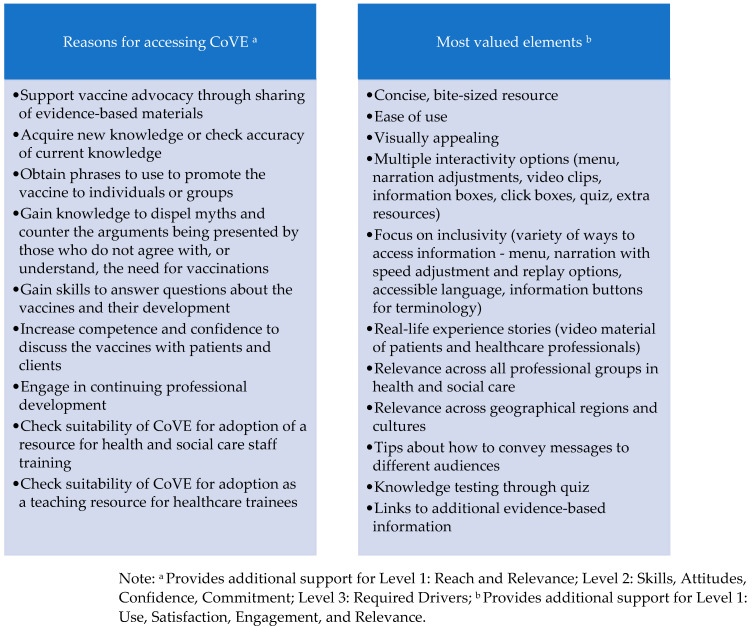
Reasons for access and most valued training elements.

**Table 1 ijerph-19-00653-t001:** Mapping of design principles to RLO design feature.

Design Principle	Learning Approach for the RLO
Multimedia	Combination of words, video, and images.
Segmenting	Eleven learner-paced segments (rather than a continuous unit).
Pre-training	Learning outcomes provided, menu provides names of the key concepts.
Modality	Contains animation and narration (in addition to on-screen text).
Coherence	Exclusion of extraneous words, pictures, and sounds through colours, boxes, and moving materials to further resources.
Redundancy	Removal of superfluous on-screen text. Inclusion of slides with visuals and audio only.
Signalling	Use of a menu, navigation buttons, and section numbering.
Spatial contiguity	Ensuring proximity of related words and images.
Temporal contiguity	Ensuring related words and images appear at the same time.
Personalisation	Text is presented in the active voice (conversational style).
Voice	Human narration of text content.

**Table 2 ijerph-19-00653-t002:** Measurement aligned with the New World Kirkpatrick Evaluation Framework.

Level (1–4) ^†^	Sub-Component	Measure	Data Collection		
			Pre-Survey	Post-Survey	Interview
1	Reach	Channel for receipt of the resource User role: healthcare professional or student Geographical region		X	X
	Use	Ease of use Helpfulness for learning Main reason for accessing Problems with use (technical, level of difficulty, context, cultural)		X	X
	Satisfaction	Overall view and rating of the resource Elements most liked Elements least liked Recommendation to others		X	X
	Engagement	View towards interactive elements (menu, narration adjustments, video clips, information boxes, click boxes, quiz, extra resources)		X	X
	Relevance	Relevance to self or others Opportunity to use the resource		X	X
2	Knowledge	Evidence of new learning	X	X	X
	Skill	Feeling equipped with useful knowledge		X	X
	Attitude	Views towards COVID-19 vaccine/change in views		X	X
	Confidence	Changes in confidence to communicate (patients or clients)	X	X	X
	Commitment	Estimated future use and resource sharing		X	X
3	Behaviour changes	User application of knowledge Reported behavioural changes		X	X X
	Required drivers	Target audiences Mechanisms for dissemination		X	X X
4	Leading indicators	Changes in user confidence Changes in user communication Resulting patient, client or general public actions Additional perceived benefits or applications		X	X X X X

^†^ Level descriptors—Level 1: Reaction; Level 2: Learning; Level 3: Transfer/Behaviour; Level 4: Results/Impact.

**Table 3 ijerph-19-00653-t003:** Mixed-methods analysis aligned with the New World Kirkpatrick Evaluation Framework.

Level (1–4) ^†^	Sub-Component	Measure ^a^	*N* (%)
**(1)** **Reaction**	Reach	Channel for receipt of the resource ^a^	
		Through employer	81 (50)
		Through educational institution	22 (13.6)
		Via professional network	35 (21.6)
		Recommended by peer/colleague	22 (13.6)
		Through digital catalogues	3 (1.8)
		Other route (e.g., family, manager)	9 (5.6)
		User ^a^	
		Health or care professional	116
		University or college students	(71.6)
		Tutor/teacher/lecturer	22 (13.6)
		General public	16 (9.9)
		Other (e.g., public health specialist/	8 (4.9)
		researcher, professional network manager)	20 (12.3)
		“in Indonesia particularly, we are struggling for vaccination today...by providing this educational package it helps health care professional to explain clearly for the patients the technical point...of vaccinations that it will make and convince people to get vaccinated.” (109)	
	Use	Easy to use	160 (98.8)
		Helpful or very helpful rating	162 (99.4)
		Problems with use (% yes)	
		No problems	152 (93.8)
		Technical issues	7 (4.3)
		Level of difficulty	1 (0.6)
		Language difficulty	0 (0.0)
		Contextual or cultural differences	1 (0.6)
		Other issues (e.g., personal device	3 (1.9)
		issue, lack of time to complete)	
		“easy to follow and informative and it wasn’t too long but I felt it covered everything that needed to be covered” (114) “we have a lot of staff who English is not their first language and I felt it was understandable and easy” (105) “a variety of ways of accessing the information” (S)	
	Satisfaction	Good or excellent rating	161 (99.9)
		Would recommend to others	160 (98.8)
		“I would say this is, I think, is the material that I was looking for. I am really impressed with this” (106) “this is very beneficial for us, our welfare. Removing the rumours about... the COVID-19” (110) “brief and to the point, but extensive extra resources giving further detail if you want it” (S)	
	Engagement	View towards interactive elements: “very interactive and engaging—information buttons to explain all the terms, text boxes to expand, images, videos, narration and additional reading. I revisit it and find more information each time” (S) “the graphics of it, the way it was quite interactive, you can click on different things... you don’t have to sit and read. You could just listen to it and that was really good” (113)	
	Relevance	Relevance to self or others: “I think this one is really timely because the level of vaccine hesitancy among nurses in the Philippines is a bit high as well” (106) “the patient experiences... I feel these are the stories that will help others understand the need more” (S) “I know a lot of my colleagues, it’s information they don’t have access to” (102) “I work in...the front line...COVID dilemmas happen every day. So, yes, I, I do believe that this information is pertinent” (112) “contain a very reliable information that we can share to the patient and convince them’ (109)	
**(2)** **Learning**	Knowledge	Pre-knowledge score ≥ 8/10	57 (35.2)
		Post-knowledge score ≥ 8/10	138 (84.6)
		Learned something new (% yes)	139 (85.8)
		“almost everything is new for me in this resource” (115) “I found the explanation of the clinical trials and the different phases quite useful ‘cause that wasn’t something I knew about and it’s where a lot information I’ve seen being spread through social media is about.” (102) “it gave me better insight into the actual client t hat I’m dealing with and all the emotions” (111)	
	Skill	Feeling equipped with useful knowledge: “now...I’m armed with new information and how to explain it to them [patients]” (106) “the learning is really related to how to present the facts...really hones in on how to communicate that knowledge I think” (114) “It would help me facilitate a conversation about COVID to people” (112)	
	Attitude	Views towards COVID-19 vaccine: “after assessing the resource it makes me more confident about the vaccines” (109) “I can imagine if somebody was very anxious, and quite sceptical. I think this this will be very good for them” (113) “It will erase their individual beliefs about the negative things about or information about COVID-19 vaccine” (107) “it strengthened my belief now that now we have to tell people the correct information” (106) “I wouldn’t say it changed my views ‘cause I was always very positive about the vaccination…but it has cemented t hem.” (105) “I manage a care company with 108 staff, 13 of those are currently refusing the vaccination. I wish to support them to gain further correct knowledge to hopefully dispel any fears and take up the vaccine” (S)	
	Confidence	Pre-confidence score ≥ 8/10	72 (44.5)
		Post-confidence score ≥ 8/10	130 (80.2)
		“now that I have this resource behind me. It [gives me] more credibility. It’s not just my opinion now” (105) “There are lot of rumours regarding the negative reaction of vaccine in my society, but by using this resource I can better explain the effectiveness of vaccine with opponents and encourage them to get vaccine.” “gave me more confidence. it was very transferable knowledge” (112) “in terms of talking to a stranger about vaccines and, you know, how they work, I’m more confident now” (101)	
	Commitment	Estimated future use and resource sharing: “I’m going to promote this material because this is relevant. There’s no such materials, I would say at the moment in the Philippines.” (106) “we’ve got some healthcare staff that are resistant...because of the propaganda...I would be quite happy to use this resource in a discussion forum with a group of staff. To enable us to have those difficult conversations really” (105) “I definitely share the package to my students.” (110) “support workers...or students... I could use that knowledge to help and support them for s ure” (103)	
**(3)** **Transfer/** **Behaviour**	Behaviour changes	User application of knowledge and reported behavioural changes: “I have already applied it. I applied it on my family because I am encouraging my parents to get vaccinated with the COVID-19 vaccine... they are afraid to get vaccinated.” (109) “I have shared this resource to all my family members, society and colleagues. I have planned to conduct awareness session at District level.” (115) “I have applied this at my workplace and shared link... among my colleagues and representatives of NGOs [non-profit organisations]. They all given me a good response.” (115)	
	Required drivers	Target audiences and mechanisms for dissemination “Maybe in the pharmacy, actually like community pharmacists when patients come in to get the medicine” (113) “I think nurses are a good place to start because they can, they can pass the knowledge onto others. Students are a good one I think because obviously they’re going into all these different places and meeting all these different people” (101) “looking at more of the health care assistants… I think I’ve found that they’re the ones who are more likely to have their own misunderstandings, which makes it harder for them to give suitable information to patients” (102) “State Governments may adopt this for sharing through Health Departments” (115) “In our place, there a lot of rural area where they don’t have much, uh, like cell phones or technology. So probably we, we can have like giving them a hard copy about this” (107) “We talked at the vaccination centre about how it might be useful for everybody to do and be part of the training process. I’ve shared it on WhatsApp and quite a few people have done it already. I also shared it in my trust with our new vaccination lead” (114)	
**(4)** **Results/** **Impact**	Leading indicators	Changes in user confidence or communication; Resulting patient, client or general public actions; Additional perceived benefits or applications: “I am about to receive a vaccine this weekend” (S) “Being able to answer the questions about how they got a vaccine available so quickly, which is the one I seem to be faced with a lot... [I am] having these conversations with the public” (102) “I’ve shared the knowledge presented in the package to my colleague and now as well as my family...they become more confidenced that this is one of our way to protect our family and our self” (109) “I have applied this knowledge and I am shocked that I have convinced each of them to take vaccine and answers their queries better and changed their view and mind about the negativity of vaccine. It is wonderful experience and I have observed that if one’s can explain better, he definitely will get the goals”. (115)	

^†^ Level descriptors—Level 1: Reaction; Level 2: Learning; Level 3: Transfer/Behaviour; Level 4: Results/Impact ^a^ Multi answer: Percentage of respondents who selected each answer option (e.g., 100% would represent that all this question’s respondents chose that option). ^a^ Quotations provided are from interviews (labelled with participant number) or Survey (labelled S).

## Data Availability

The CoVE package is open access on HELM Open: https://www.nottingham.ac.uk/helmopen/rlos/practice-learning/public-health/CoVE/ (accessed on 17 December 2021). The evaluation data presented in this study are available on request from the corresponding author. The data are not currently publicly available due to risk of participant identification.

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
