# Peer review of "COVID-19 Vaccine Education (CoVE) for Health and Care Workers to Facilitate Global Promotion of the COVID-19 Vaccines"

_ijerph, 2022, doi:10.3390/ijerph19020653_

Round 1

Reviewer 1 Report

The manuscript entitled “COVID-19 Vaccine Education (CoVE) for health and care workers to facilitate global promotion of the COVID-19 vaccines” demonstrates the study employing an internationally relevant, interactive multimedia training resource called COVID-19 Vaccine Education (CoVE), which was developed using ASPIRE methodology. The work is presented in approachable, but scientific manner. The manuscript is technically appropriate, and the data support the conclusions. There are few minor suggestions to be proposed to enhance the manuscript:

Line 58: “less likely to take up vaccination [4][18, 19].”

Suggestion: Please combine the citation in one square bracket.

Line 127: “ASPIRE has a 6-step process [35][43] including:”

Suggestion: Please combine the citation in one square bracket.

Line 296: “Evaluation aligned with the four levels of the New World Kirkpatrick Model [36, 37] [44],”.

Suggestion: Please combine the citation in one square bracket.

Based on article evaluation, I recommend Minor Revision of this manuscript.

Author Response

Dear Editor and Reviewers,

Many thanks for taking the time to review our manuscript. Please find our changes highlighted in the uploaded manuscript, and our responses uploaded as a separate document to address all reviewer comments. We appreciate your input and believe this has improved the quality of the manuscript. We look forward to hearing from you.

Kind regards,

Holly Blake (corresponding author)

Reviewer 2 Report

1) Is the learning objective clear and do all sections of the RLO support it?

Is Revision Required?       Yes â–¡              No â–ª

The aim of the study was clearly defined.
It was to seek answers to 2 questions:
1) does the digital training package developed by the authors increase user knowledge and confidence in promoting the COVID-19 vaccine?
2) does the digital training package lead to changes in the behaviour of users with regard to promotion of the vaccine?
The paper is structured according to the requirements of a scientific journal.

2) Is the content factually correct?

Is Revision Required?       Yes â–¡              No â–ª

The reviewed article presents the methodology of creating training materials supporting education on COVID-19 for the purposes of healthcare. This is a very important topic that the whole world is facing at the moment.

3) Is the text well written in short, clear, sentences?

Is Revision Required?       Yes â–¡              No â–ª

The text is written in clear sentences.

4) Does the glossary cover all the terms required for a general audience?

Is Revision Required?       Yes â–¡              No â–ª

The glossary cover the terms required for a general audience.

5) Is the structure and sequence of information helpful?

Is Revision Required?       Yes â–¡              No â–ª

The topic bridges the gap in the field of educational methods and the art of persuading people to vaccinate the entire world population. It shows a methodical way to create educational materials in the healthcare sector.

6) Are the suggestions/examples for images/animations/video appropriate?

Is Revision Required?       Yes â–¡              No â–ª

The pictures and animations in the article complement the completeness of the research.

7) Is sufficient interaction proposed to support active learning?

Is Revision Required?       Yes â–¡              No â–ª

The article shows a method of creating teaching materials used in the healthcare sector to convince patients to vaccinate. At the same time, medical personnel is involved in each stage of creating these materials as a test group.

8) Will the assessments measure attainment of the learning objective?

Is Revision Required?       Yes â–¡              No â–ª

The methodological side does not raise my reservations. The research is carried out in accordance with the art of conducting such research in the subject matter discussed in the article.

9) Are the keywords appropriate? Are others needed?

Is Revision Required?       Yes â–¡              No â–ª

The keywords represent the terms used in the paper.

10) Are the suggested links OK? Are there others that you could suggest?

Is Revision Required?       Yes â–¡              No â–ª

The literature cited in the work corresponds to the current state of research in the field of the discussed issues. It should be emphasized that the literature is quoted mainly from the last two years, which also indicates the topicality of the analyzed research topic.

11) Have you discussed your review with the authors? Nature of communication (eg face-to-face, e-mail etc)

No, I haven't.

Additional comments or continuations of above sections.

The conclusions are correctly formulated and result from the conducted analyzes. The adopted research goal was achieved at a satisfactory level.

The paper is well prepared. I suggest adopting it as it stands.

Author Response

(The authors gave the same response as above.)

Reviewer 3 Report

The manuscript by Blake et al., entitled "COVID-19 Vaccine Education (CoVE) for health and care workers to facilitate global promotion of the COVID-19 vaccines," describes the development of an on-line educational program to provide health care workers with information and training regarding the use of COVID-19 vaccination. This is a well-written and presented manuscript and describes an important avenue for dissemination of vaccine information to health care workers. As the investigators indicate, health care workers are at the front line of vaccine promotion and distribution to the general public and thus, have a crucial function in promoting public health measures. Below are a few comments that may be considered prior to publication of the manuscript.

  1. Lines 161-66: Ethnic minority groups were overrepresented in group 2. Please explain the reasoning behind this choice.
  2. Lines 208-11: How was this group of experts chosen? Does the selection process have a potential impact on outcome?
  3. Lines 256-60: Is inclusion of representation from seven countries sufficient to "establish relevance and appropriateness of content across cultures and geographical regions." Maybe this statement should be reworded.
  4. Lines 320-21: Should there be some concern that the selection process to identify interview participants is introducing a bias? It is likely that health care workers opting to participate in this survey have a more favorable view of this training process.
  5. Line 325: As stated in the manuscript, interviews took place shortly after participants had completed the training. Can you please be more specific (hours, days, weeks?). In addition, it might be useful to conduct a follow-up assessment, to evaluate if these outcomes still applied after health care workers were likely exposed to varying COVID-19 vaccine information stemming from friends/family, the media, and social network platforms.
  6. How was sample size determined (e.g., do 15 interviewees and 162 online participants truly represent specific needs of all of these cultures and regions)?

Author Response

(The authors gave the same response as above.)
